

# CINmetrics: an R package for analyzing copy number aberrations as a measure of chromosomal instability

Vishal H. Oza[1], Jennifer L. Fisher[1], Roshan Darji[2] and Brittany N. Lasseigne[1]

[1] Department of Cell, Developmental and Integrative Biology, University of Alabama - Birmingham, Birmingham, AL, United States of America
[2] HudsonAlpha Institute for Biotechnology, Huntsville, AL, United States of America

## ABSTRACT

Genomic instability is an important hallmark of cancer and more recently has been identified in others like neurodegenrative diseases. Chromosomal instability, as a measure of genomic instability, has been used to characterize clinical and biological phenotypes associated with these diseases by measuring structural and numerical chromosomal alterations. There have been multiple chromosomal instability scores developed across many studies in the literature; however, these scores have not been compared because of the lack of a single tool available to calculate and facilitate these various metrics. Here, we provide an R package CINmetrics, that calculates six different chromosomal instability scores and allows direct comparison between them. We also demonstrate how these scores differ by applying CINmetrics to breast cancer data from The Cancer Genome Atlas (TCGA). The package is available on CRAN at https://cran.rproject.org/package=CINmetrics and on GitHub at https://github.com/lasseignelab/CINmetrics.

## INTRODUCTION

Genomic instability, one of the hallmarks of cancer and aging, is measured in many forms such as chromosomal instability, microsatellite instability, and instability characterized by increased frequency of base-pair mutations (*Bakhoum & Cantley, 2018*; *Pikor et al., 2013*; *Negrini, Gorgoulis & Halazonetis, 2010*; *López-Otín et al., 2013*). Particularly, chromosomal instability (CIN) is associated with cancer progression, tumor immunity, and inflammation (*Pikor et al., 2013*; *Bach, Zhang & Sood, 2019*). Recently, CIN has been shown to contribute to diseases other than cancer, including neurodegenerative diseases (*Hou et al., 2017*; *Yurov, Vorsanova & Iourov, 2019*).

CIN is broadly defined as the change in number and structure of chromosomes (*Vargas-Rondón, Villegas & Rondón-Lagos, 2017*). Since CIN involves simultaneous and ongoing copy number changes, it is a dynamic phenotype that leads to intratumoral heterogeneity. In many published studies, CIN has been measured in proxy by capturing errors during anaphase segregation in tumor specimens or indirectly by measuring

Corresponding author
Brittany N. Lasseigne,
bnp0001@uab.edu

numerical and structural chromosomal alterations across cell populations (segmental aneuploidy) (*Baumbusch et al., 2013*; *Davison et al., 2014*; *Roylance et al., 2011*; *Bonnet et al., 2012*). However, they do not measure the state or rate of chromosomal change, which is a limitation of array based and comparative genomic hybridization (CGH) methods (*Geigl et al., 2008*; *Lepage et al., 2019*). There have been various CIN scores developed across multiple studies involving different cancers which calculate numerical and structural alterations in the chromosome (*McGranahan et al., 2012*). The differences in calculation of these scores have been associated with different clinical and biological phenotypes (*Baumbusch et al., 2013*; *Davison et al., 2014*; *Roylance et al., 2011*; *Bonnet et al., 2012*); however, there has been no systematic comparison of different CIN scores and how they vary across and within different cancers. The primary reason being lack of availability of computational framework to calculate these CIN scores. While other packages are available to calculate chromosomal instability (*Song et al., 2017*), they are limited to a single chromosomal instability score and do not provide a framework to calculate and compare other CIN scores. Here, we provide an R package that provides a unified framework to calculate multiple CIN metrics to quantify segmental aneuploidy accumulations in a sample on same dataset. This package will accelerate chromosomal instability studies by facilitating score comparisons across cancers or other diseases.

## METHODS

The chromosomal instability metrics were mined from the cancer literature and implemented as functions in our CINmetrics R package, based on their ability to detect either structural, numerical, or whole genome instability. The six functions (*tai, taiModified, cna, countingBreakPoints, countingBaseSegments, fga*) are outlined below based on the similarity of the algorithms used to calculate them.

### Total aberration index *(tai)* and modified total aberration index *(taiModified)*

Total Aberration Index (TAI) was proposed by *Baumbusch et al. (2013)* to measure the genomic aberrations in serous ovarian cancers. TAI calculates absolute area under the curve for a copy number segment profile generated by piecewise constant fitting (PCF) algorithm (*Baumbusch et al., 2008*). Biologically, TAI can be interpreted as absolute deviation from the normal copy number state averaged over all genomic locations. TAI provides a numerical measure in terms of both prevalence as well as the genomic size of copy number variations in tumors. One of the limitations of TAI is that since it was designed for studying advanced stage ovarian tumors, short aberrations found in early stage tumors have low impact on TAI. Therefore, TAI should be used to study the global scale genomic disorganization most likely to occur in late stage tumors.

   *tai* implemented in CINmetrics takes into account only those sample values that are in aberrant copy number state, *i.e.,* has a mean segment value of less than or equal to $-0.2$ and greater than or equal to $+0.2$, without taking into account whether it is a deletion or

amplification.

$$Total\ Aberration\ Index = \frac{\sum_{i=1}^{R} d_i \cdot |y_{S_i}|}{\sum_{i=1}^{R} d_i} \tag{1}$$

where $y_{S_i} \leq -0.2$ and $y_{S_i} \geq +0.2$ represents the mean segment value, $d_i$ represents the segment length, and $R$ represents the total number of segments. Alternatively, *taiModfied* takes into account all the mean segment values and thus preserves the "directionality" of the score. In other words, *taiModified* retains the "directionality" of amplification or deletion. Negative modified tai value means there are more large deletions in the sample since the negative segment mean values are driving the metric.

$$Modified\ Total\ Aberration\ Index = \frac{\sum_{i=1}^{R} d_i \cdot y_{S_i}}{\sum_{i=1}^{R} d_i} \tag{2}$$

where $y_{S_i}$ represents the mean segment value, $d_i$ represents the segment length, and $R$ represents the total number of segments.

## Copy number abnormality *(CNA)* and number of break points *(countingBreakPoints)*

Copy number abnormality (CNA) was developed by *Davison et al. (2014)* for studying aneuploidy in superficial gastroesophageal adenocarcinoma. An individual CNA is defined as the segment with copy number outside the predefined range of 1.7 to 2.3 where two indicates no loss or gain (assuming that the tumor is diploid) as determined by the Partek segmentation algorithm (*Grayson & Aune, 2011*). Total CNA for the sample can thus be defined as total number of individual CNAs. CNA represents a measure of segmental aneuploidy. *cna* implemented in CINmetrics is similar except we define individual CNA as the segment with copy number less than or equal to $-0.2$ and greater than or equal to $+0.2$ with segment mean of 0 indicating no loss or gain. We chose $\pm 0.2$ as a conservative cutoff for TCGA data as described in *Laddha et al. (2014)*. The users can modify the cutoff by modifying *segmentMean* parameter.

$$Total\ Copy\ Number\ Abnormality = \sum_{i=1}^{R} n_i \tag{3}$$

where $n_i$ represents number of segments with $y_{S_i} \leq -0.2$ and $y_{S_i} \geq +0.2$, $R$ represents the total number of segments with the minimum segment length $d_i$ greater than or equal to 10.

*countingBreakPoints* is similar to the total breakpoints implemented in *Lee et al. (2011)*. Segments with mean less than or equal to $-0.2$ and greater than or equal to $+0.2$ and that contain a number of probes above the user defined threshold, are counted and then the value is doubled to account for 3′ and 5′ ends. This metric yields similar relative distribution results to *cna*, however there is no minimum segment length ($d_i$) filter so the actual metric values differ (they will be higher for *countingBreakPoints*).

$$Number\ of\ Break\ Points = \sum_{i=1}^{R} (n_i \cdot 2) \tag{4}$$

where $n_i$ represents number of segments with $y_{S_i} \leq -0.2$ and $y_{S_i} \geq +0.2$ and $R$ represents the total number of segments.

## Counting altered base segments *(countingBaseSegments)* and the fraction of the genome altered *(fga)*

Counting altered base segments and fraction of the genome altered are modified implementations of the Genome Instability Index (GII) as described in *Chin et al. (2007)*. The GII was computed in two different ways, both based on calculating common regions of alteration (CRA). These approaches show high concordance.

$$Number\ of\ Altered\ Bases = \sum_{i=1}^{R} d_i \tag{5}$$

where $d_i$ represents length of segments with $y_{S_i} \leq -0.2$ and $y_{S_i} \geq +0.2$ and $R$ represents the total number of segments.

*fga* implemented in our package is based on identifying CRAs as fraction of the genome altered. Therefore, the *fga* values are normalized by dividing it by the length of the genome covered. *countingBaseSegments* on the other hand calculates the CRAs.

$$Fraction\ Genome\ Altered = \frac{\sum_{i=1}^{R} d_i}{G} \tag{6}$$

where $d_i$ represents length of segments with $y_{S_i} \leq -0.2$ and $y_{S_i} \geq +0.2$, $G$ represents genome length covered, and $R$ represents the total number of segments. The default value is calculated by adding length of each probe on Affymetrix 6.0 array file (snp6.na35.remap.hg38.subset.txt.gz) found here (*GDC, 2023*) after excluding the sex chromosomes. One important difference to note is that in the original GII calculations, the algorithm merges the overlapping regions between samples, whereas *fga* and *countingBaseSegements* implemented in CINmetrics package do not merge overlapping regions, since TCGA segments are not overlapping. However, if the user is using their own segmentation algorithm, they should make sure that the segments are not overlapping otherwise it will lead to erroneous results.

## RESULTS

We used harmonized masked copy number segment data for breast cancer (BRCA) from The Cancer Genome Atlas (*Cancer Genome Atlas Network, 2012*; *Cerami et al., 2012*) to visualize and compare the chromosomal instability metrics implemented in the CINmetrics package. We chose the breast cancer data as it has been shown to exhibit chromosomal instability and thus provides a robust dataset for applying CINmetrics *e.g.*, (*Duijf et al., 2019*; *Voutsadakis, 2021*). Figure 1A shows the distribution of CINmetrics in BRCA data for normal and tumor samples. The metrics have been $\log_{10}$ scaled to allow for comparison between them. *cna*, *countingBreakPoints*, *fga*, and *countingBaseSegments* show an overall pattern of increased genomic instability in tumor samples compared to normal. However, the difference in mean and standard deviation between the two classes(normal and tumor) is very different between these metrics. *tai* and *taiModified* scores show greater overlap

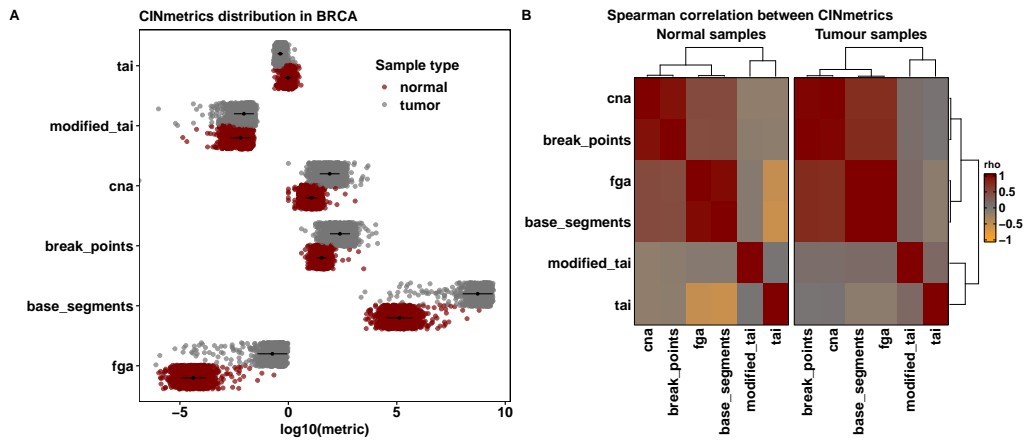

**Figure 1** CINmetrics applied to the BRCA dataset from TCGA, (A) the distribution of metrics between normal (red) and tumor (grey) samples, where the black dot indicates the mean and the black line indicates the standard deviation and (B) heatmap of the spearman correlation and complete linkage clustering of the metrics in normal and tumor samples.

between normal and tumor samples compared to other metrics. As mentioned earlier, *tai* and *taiModified* are best suited for late stage cancers (*Baumbusch et al., 2013*), thus should be used as a measure for studying overall genomic disorganization in individual patients with advanced tumors and not as a measure of genomic instability comparison between normal and tumor samples, as we further demonstrate here.

To further understand and characterize the relationship between various metrics implemented in CINmetrics, we performed spearman correlation (*Spearman, 1904*), followed by complete linkage clustering (*Vijaya, Sharma & Batra, 2019*) as shown in Fig. 1B. This clustering further demonstrated that *cna*, *countingBreakPoints*, *fga*, and *countingBaseSegments* are more similar and therefore highly correlated compared to *tai* and *taiModified*. Furthermore, this relationship is preserved in both normal and tumor samples indicating the four metrics show consistent results and can be used for comparing genomic instability between the two conditions.

We also looked at how the metrics are affected by potential confounders such as tumor purity (defined as the fraction of cancerous cells in tumor samples) and ploidy levels in tumor samples in the BRCA dataset. We obtained the purity and ploidy data for BRCA calculated using ABSOLUTE algorithm (*Carter et al., 2012*) from the NIH Genomic Data Commons Portal (*The Pan-Cancer Atlas, 2022*). ABSOLUTE jointly infers tumor purity and ploidy levels from allele-specific copy number levels from a large sample collection and precomputed models of recurrent cancer karyotypes (*Carter et al., 2012*).

For purity, the purity score had a range between 0 and 1 for each sample, with 1 being the highest purity (Fig. 2A). We divided the score in four quantiles and plotted the density of the samples in each quantile against the CINmetrics scores (Fig. 2). All CINmetrics had relatively lower scores for samples with less purity (1st Quantile). Interestingly, *tai* showed a distinct increase in the score with the increase in purity of samples. These can be due

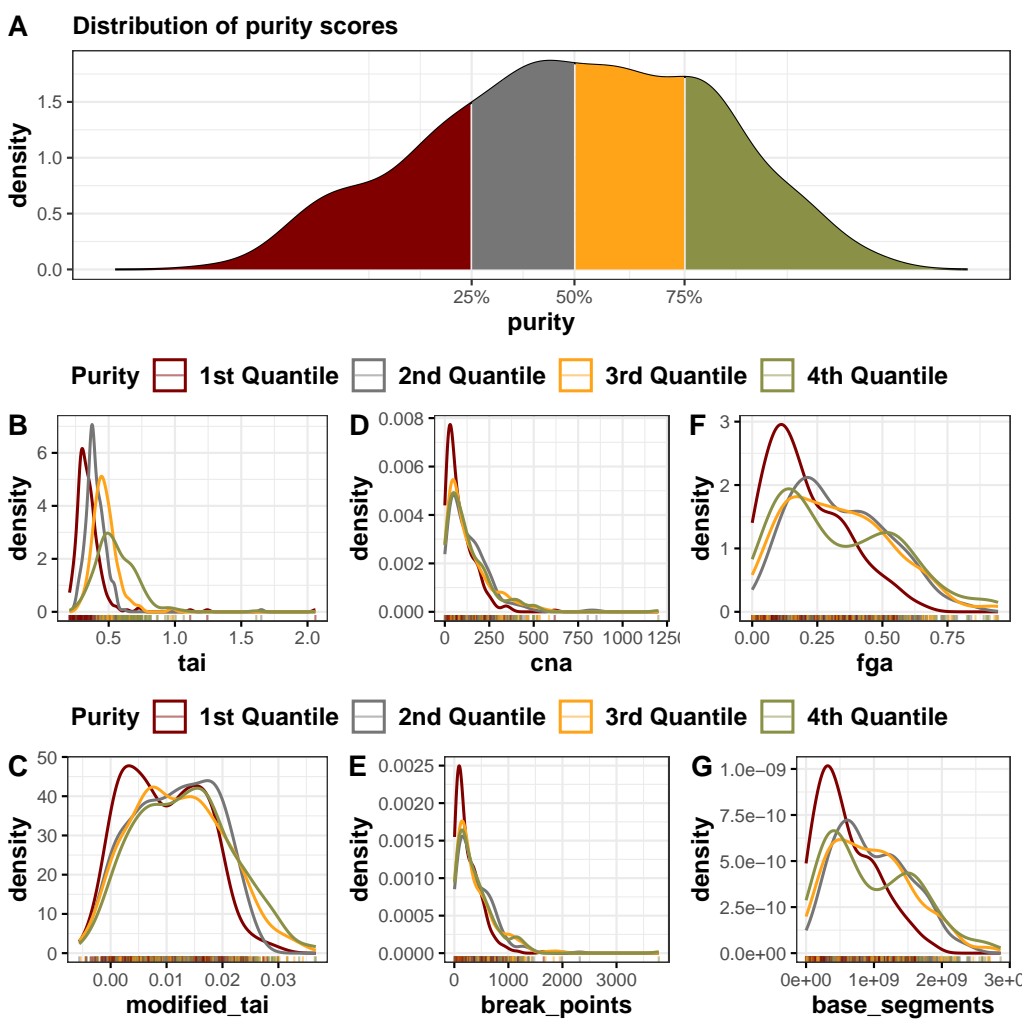

**Figure 2** Distribution of sample purity scores and CINmetrics applied to BRCA dataset from TCGA compared to sample purity, (A) sample purity score quantile distribution (B) *tai*, (C) *taiModified*, (D) *cna*, (E) *countingBreakPoints*, (F) *fga*, and (G) *countingBaseSegments*.

to more purer samples having higher segmental aneuploidy and therefore more segments having mean segment value of greater than 0.2. *taiModified* showed the same distribution across the four quantiles.

For ploidy, we looked at the density of samples with different ploidy numbers against the CINmetrics scores (Fig. 3). Samples that were diploid (2n) had the lowest score across *cna*, *countingBreakPoints*, *fga*, and *countingBaseSegments*; however not in *tai* and *taiModified*. *cna* and *countingBreakPoints* were developed to study aneuploidy (*Davison et al., 2014*; *Lee et al., 2011*), and they show higher scores corresponding to higher ploidy levels.

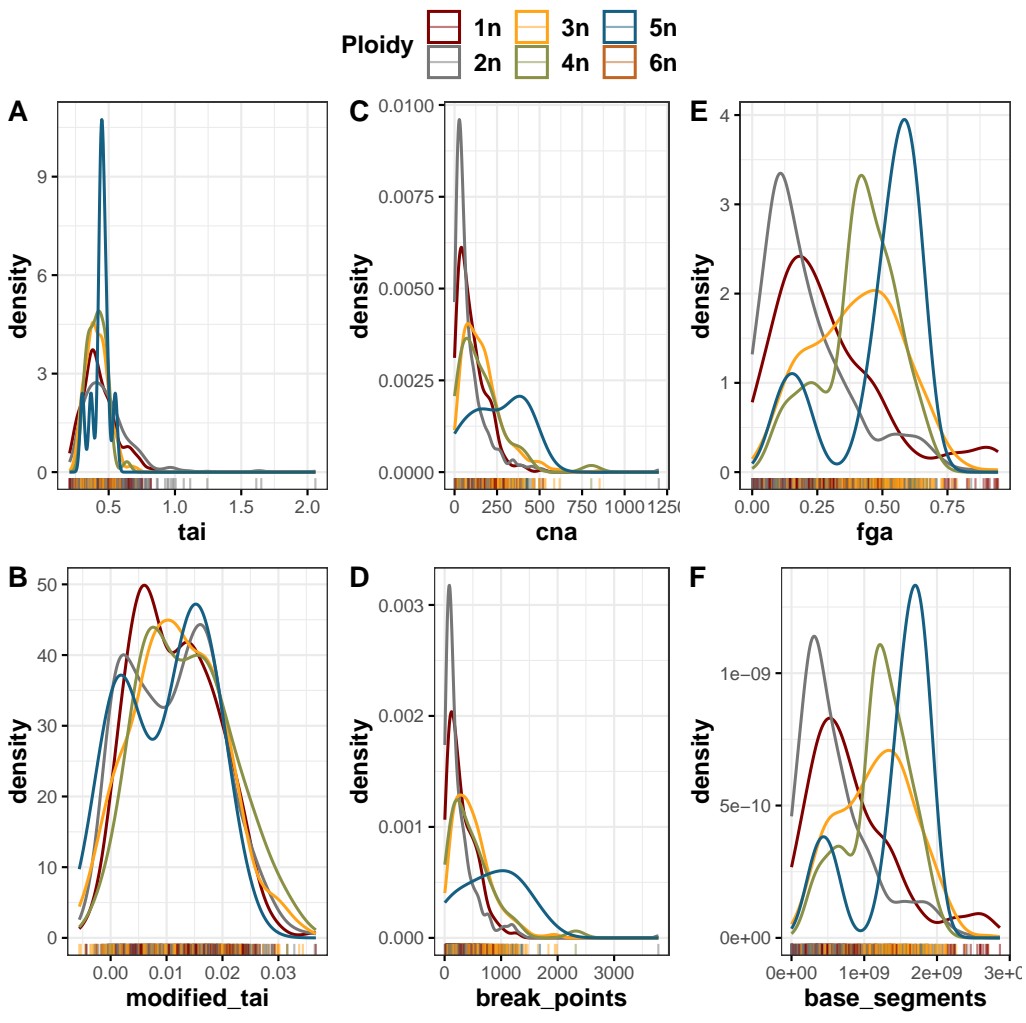

**Figure 3** Distribution of CINmetrics applied to the BRCA dataset from TCGA compared to sample ploidy levels, (A) *tai*, (B) *taiModified*, (C) *cna*, (D) *countingBreakPoints*, (E) *fga*, and (F) *countingBaseSegments*.

## CONCLUSIONS

CIN has been one of the most important factors in understanding disease etiology and progression in cancer (*Pikor et al., 2013*; *Bach, Zhang & Sood, 2019*) and is becoming increasingly recognized for others like neurodegenerative diseases (*Hou et al., 2017*; *Yurov, Vorsanova & Iourov, 2019*). Also, genomic instability has been associated with biological variables such as sex, age, and tissue (*Fischer & Riddle, 2018*). Numerous methods have been developed to quantitate and characterize the role of chromosomal instability in specific cancers, however, lack of comprehensive tools that calculates these metrics has limited direct comparison between them. Here, we have collected chromosomal instability metrics from the literature and provide them as an R package and associated vignette that allows for reproducible calculations and comparisons. Further, we used BRCA data from

The Cancer Genome Atlas to show how the metrics relate to each other. One limitation of these scores is that if the aneuploidy arose at the outset of cancer, these scores might give high scores even in absence of CIN. The advent of next generation sequencing and single cell technologies allows for better measurement of the state of chromosomal change as well as cell-to-cell variability which is often masked in traditional chromosomal instability scores (*Geigl et al., 2008*; *Lepage et al., 2019*). Another limitation of the package is that it relies on array based intensity scores for the calculations of copy number variation and therefore cannot be used with next generation sequencing data. This package nevertheless, provides a useful framework to better characterize and understand genomic instability in cancer.

# ACKNOWLEDGEMENTS

We would like to thank Tabea M. Soelter and other members of the Lasseigne Lab for their valuable input.

## Funding

This work was supported by the NIH R00HG009678 (to Brittany N. Lasseigne), NIH R00HG009678-04S1 (to Brittany N. Lasseigne ), UAB AMC21 (to Jennifer L. Fisher), and UAB Startup Fund Support (to Brittany N. Lasseigne). The funders had no role in study design, data collection and analysis, decision to publish, or preparation of the manuscript.

## Grant Disclosures

The following grant information was disclosed by the authors:
The NIH R00HG009678.
NIH R00HG009678-04S1.
UAB AMC21.
UAB Startup Fund.

## Competing Interests

The authors declare there are no competing interests.

## Author Contributions

- Vishal H. Oza conceived and designed the experiments, performed the experiments, analyzed the data, prepared figures and/or tables, authored or reviewed drafts of the article, and approved the final draft.
- Jennifer L. Fisher conceived and designed the experiments, authored or reviewed drafts of the article, and approved the final draft.
- Roshan Darji conceived and designed the experiments, performed the experiments, authored or reviewed drafts of the article, and approved the final draft.
- Brittany N. Lasseigne conceived and designed the experiments, authored or reviewed drafts of the article, and approved the final draft.

## Data Availability

The code is available on CRAN: https://cran.r-project.org/package=CINmetrics.

The data and code are available at GitHub and Zenodo: https://github.com/lasseignelab/CINmetrics.

Vishal H. Oza. (2023). lasseignelab/CINmetrics: inital release of CINmetrics R package repo (v1.1.0). Zenodo. https://doi.org/10.5281/zenodo.7750898.

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
