# Peer review of "CINmetrics: an R package for analyzing copy number aberrations as a measure of chromosomal instability"

_PeerJ, doi:10.7717/peerj.15244_

## Round 0.1 · original submission · Major Revisions

Thank you for submitting your manuscript to peerJ. The manuscript reads very well and is indeed an important area of research.

After receiving comments from all reviewers, it is clear that the manuscript requires major revisions before it can be accepted. In particular, I want to highlight the comments of reviewer 2 (below), who points out that the manuscript needs more details to support its claims - the authors need to provide the information requested by reviewer 2 to support the integrity of their workflow.

·

Basic reporting

In this manuscript, the authors proposed an integrated tool for investigation of chromosome stability. They have tested the performance of they tool and successfully demonstrated the usefulness. Since there are no other tools that allow people to perform this kind of integrated analyses, the development of the present tool and the publication of the present manuscript is acceptable.

Experimental design

There are no experimental design for this manuscript since it reports the new tool.

Validity of the findings

There are no findings in this paper since it reports the new tool.

Additional comments

No additional comments.

Reviewer 2 ·

Basic reporting

1) The authors should explicitly distinguish CIN from aneuploidy as they are often conflated (perhaps between lines 29-32). Consider providing the caveat that CINmetrics provides proxy measures of CIN by quantifying the accumulated aneuploid copy number alterations in a sample, which does not detect ongoing CIN. If the aneuploidy arose at the outset of the cancer, there may be, in fact, aneuploidy without CIN which would score high by these measures. Review and consider citing PMID: 18192061.
2) Some of the equations appear to be missing absolute value backets found in the LateX code in the GitHub repository. Please review the notation in the manuscript. For example, the equations (1) and (2) appear identical, but (1) should have absolute value brackets.
3) Line 127: Please clarify what the purity score represents and consider displaying a distribution of purity scores so the quantiles can be interpreted.
4) Most new data being generated is short-read NGS. Consider describing how the package could be applied to short-read NGS in the manuscript or in the package vignette.
5) Consider removing the translucent white fill on the density plots (and/or change the color scheme) as they tend to obscure the lighter distributions.
6) The size of the axis/legend text and color code are small and difficult to see (particularly the small yellow dot in Fig. 1A).

Experimental design

7) Briefly clarify the meaning of “directionality” and why directionality matters. Relatedly, how should negative modified tai be interpreted in the distribution in Figure 2B?
8) Equation (4) indicates that breakpoints score is exactly double Equation (3) cna score. It seems redundant to include both measures. As expected the distributions in 2C and 2D are almost identical. However, the highest data point in Figure 2C is less than 1250 and in Figure 2D it is greater than 3500—how can this be explained?
9) Please clarify what default value you are referring to on line 96 and how it is calculated from the length of Affymetrix probes (or cite). (minor typo: Affymetrix, not Affeymetrix)
10) From lines 97-100, please clarify or expand on the significance of whether a measure merges non-overlapping regions between samples or not. This was difficult to understand.

Validity of the findings

11) Lines 111-115: “tai and taiModified do not capture this global pattern of difference between normal and…”.
There is small overlap between tai, can, and break_points for normal vs tumor. Thus saying tai doesn’t capture it appears unwarranted—it partially captures a difference at least as well as these other two measures. No evidence is provided to the statement that these measures are best suited for late-stage cancers—additional data should be provided, or cited, or consider removing this claim. Consider statistical comparisons in Figure 1A.
12) Lines 130-132: The manuscript states that TAI and modified TAI show increased scores in samples with higher purity and that this means purer samples have more chromosomal instability. This interpretation is confounded. A more likely explanation is that purer tumor samples have a greater likelihood of their aneuploid copy number ratios surpassing the threshold of ±0.2 used in CINmetrics. This gets back (1) on distinguishing ongoing chromosomal instability from detectable aneuploid copy number alterations.

Additional comments

The authors have developed an R package that assembles measures of CNA burden and aneuploidy used in the literature, which will be of great interest and convenience to those wanting to repeat these quantitative analyses in TCGA or other data. This is highly valuable in that it can provide easy access to comparative analysis of various published scores. It could be improved in three areas:
1) Most of the analyses quantify aneuploidy (the number of abnormal chromosomes or segments in a tumor) rather than CIN (the rate of ongoing change of chromosomes. The package should more correctly be called “CNAmetrics”.
2) Other points as above.

·

Basic reporting

In this study, Lasseigne and co-workers showed the unified approach (CINmetrics) that calculates and combine different chromosomal instability metrics. All the metrics were tested on the harmonized masked copy number segment data for breast cancer (BRCA) from The Cancer Genome Atlas and six different functions (tai, taiModified, cna, countingBreakPoints, countingBaseSegments, fga) were implemented based on the similarity of the algorithms.

A well-written and well-explained article. Clear and unambiguous, professional English used throughout. Literature references, sufficient field background/context provided.The figures' legends could be clear.

Experimental design

Original primary research within Aims and Scope of the journal. Research question well defined, relevant & meaningful. It is stated how research fills an identified knowledge gap. Indeed, this article suggested very useful tool to identify and compare different chromosomal instability scores, although the methods need elaboration for more clarity.

Validity of the findings

All underlying data have been provided; they are robust, statistically sound, & controlled. Conclusions are well stated, linked to original research question & limited to supporting results.

Additional comments

A well-written and well-explained article. The figures' legends could be clear. the methods need elaboration for more clarity. Line 104 says that breast cancer data was chosen for this work, is this also tested on other kind of cancers? If not, it will be interesting to see the application of CINmetrics on other caner type (for instance Lung cancer)
Please explain, if these metrics are affected by any other variables like gender or infection.

---

## Round 0.2 · Minor Revisions

You can see there is still some minor suggestion such as CRAN searching. Please fix those problems as appropriate.

Reviewer 2 ·

Basic reporting

We thank the authors for addressing the issues raised in basic reporting. The new text in lines 21-32 says "capturing errors during anaphase segregation in tumor specimens (polyploidy)"... but errors in anaphase segregation commonly lead to small numbers of chromosome errors, not polyploidy which is changes in entire chromosome sets.

Experimental design

The issues raised have been addressed except #10 by initial numbering.

10) They authors revised the text that says that one algorithm merges and two do not merge. We understand that one algorithm does and two do not merge, but, it is still unclear what the quantitative significance is of merging non-overlapping regions. i.e. why is this important? Does merging and non-merging algorithms simply give different results? Is one over-estimating CNA and the other expected to underestimate? Or, if it is a rather minor detail that impacts none of the above, why is this algorithmic detail important enough to highlight in the text at all?

Validity of the findings

These issues have been addressed, thank you.

Additional comments

Although the package is on CRAN, it seems both unfortunate and unsatisfying that a package that measures CNA burden claims to measure CIN, which is properly expressed as a rate of change of CNA burden. Some textual clarifications improve the manuscript. However, I find the title which includes both "CINmetrics" and "chromosomal instability analysis" to be doubly misleading as the authors appear to acknowledge the manuscript describes the measure of CNA and not CIN.

---

## Round 0.3 · accepted · Accept

Thanks to the author for modifying the manuscript based on the comments.